# Incidence and predictors of mortality among persons receiving second-line tuberculosis treatment in sub-Saharan Africa: A meta-analysis of 43 cohort studies

**Dumessa Edessa**[1]*, **Fuad Adem**[1], **Bisrat Hagos**[2], **Mekonnen Sisay**[3]

**1** Department of Clinical Pharmacy, School of Pharmacy, College of Health and Medical Sciences, Haramaya University, Harar, Ethiopia, **2** School of Pharmacy, College of Health and Medical Sciences, Haramaya University, Harar, Ethiopia, **3** Department of Pharmacology and Toxicology, School of Pharmacy, College of Health and Medical Sciences, Haramaya University, Harar, Ethiopia

* jaarraa444@yahoo.com

**Data Availability Statement:** All relevant data are within the manuscript and its Supporting Information files.

## Abstract

### Background

Drug resistance remains from among the most feared public health threats that commonly challenges tuberculosis treatment success. Since 2010, there have been rapid evolution and advances to second-line anti-tuberculosis treatments (SLD). However, evidence on impacts of these advances on incidence of mortality are scarce and conflicting. Estimating the number of people died from any cause during the follow-up period of SLD as the incidence proportion of all-cause mortality is the most informative way of appraising the drug-resistant tuberculosis treatment outcome. We thus aimed to estimate the pooled incidence of mortality and its predictors among persons receiving the SLD in sub-Saharan Africa.

### Methods

We systematically identified relevant studies published between January, 2010 and March, 2020, by searching PubMed/MEDLINE, EMBASE, SCOPUS, Cochrane library, Google scholar, and Health Technology Assessment. Eligible English-language publications reported on death and/or its predictors among persons receiving SLD, but those publications that reported death among persons treated for extensively drug-resistant tuberculosis were excluded. Study features, patients' clinical characteristics, and incidence and/or predictors of mortality were extracted and pooled for effect sizes employing a random-effects model. The pooled incidence of mortality was estimated as percentage rate while risks of the individual predictors were appraised based on their independent associations with the mortality outcome.

### Results

A total of 43 studies were reviewed that revealed 31,525 patients and 4,976 deaths. The pooled incidence of mortality was 17% (95% CI: 15%-18%; $I^2$ = 91.40; P = 0.00). The studies

**Funding:** The authors received no specific funding for this work.

**Competing interests:** The authors have declared that no competing interests exist.

**Abbreviations:** ART, Antiretroviral therapy; CI, Confidence Interval; HIV, Human Immunodeficiency Virus; JBI, Joanna Briggs Institute; MDR-TB, Multidrug-resistant Tuberculosis; PRISMA, Preferred Reporting Items for Systematic Review and Meta-Analysis; RR, Risk Ratio; SLD, second-line anti-tuberculosis drugs; SSA, sub-Saharan Africa; TB, Tuberculosis; WHO, World Health Organization.

used varied models in identifying predictors of mortality. They found diagnoses of clinical conditions (RR: 2.36; 95% CI: 1.82–3.05); excessive substance use (RR: 2.56; 95% CI: 1.78–3.67); HIV and other comorbidities (RR: 1.96; 95% CI: 1.65–2.32); resistance to SLD (RR: 1.75; 95% CI: 1.37–2.23); and male sex (RR: 1.82; 95% CI: 1.35–2.44) as consistent predictors of the mortality. Few individual studies also reported an increased incidence of mortality among persons initiated with the SLD after a month delay (RR: 1.59; 95% CI: 0.98–2.60) and those persons with history of tuberculosis (RR: 1.21; 95% CI: 1.12–1.32).

## Conclusions

We found about one in six persons who received SLD in sub-Saharan Africa had died in the last decade. This incidence of mortality among the drug-resistant tuberculosis patients in the sub-Saharan Africa mirrors the global average. Nevertheless, it was considerably high among the patients who had comorbidities; who were diagnosed with other clinical conditions; who had resistance to SLD; who were males and substance users. Therefore, modified measures involving shorter SLD regimens fortified with newer or repurposed drugs, differentiated care approaches, and support of substance use rehabilitation programs can help improve the treatment outcome of persons with the drug-resistant tuberculosis.

## Trial registration number

CRD42020160473; PROSPERO

## Introduction

Antimicrobial resistance to *mycobacterium* tuberculosis (TB) remains from among the most feared public health threats that commonly challenges the TB treatment success [1]. In 2019, a total of 206,030 people across the world were detected and notified to have drug-resistant TB (DR-TB), with 177,099 of them enrolled for receiving treatments [2]. According to the World Health Organization's (WHO) global estimate in 2017, from among the 558,000 people predicted to be infected with DR-TB, only 186,883 them were detected [2, 3]. This indicated that more than half of the DR-TB cases are left undetected, and a large number of these missed cases are likely to be in resource-limited settings. On top of this, treatment regimens received by persons with DR-TB are relatively complex, prolonged, costly, and associated with multiple toxicities that may lead to difficulties to complete the entire dosages [4]. The Global TB report of 2020 pertaining to a 2017 cohort of DR-TB patients indicated that 57% them completed the treatments successfully while 15% of them died, 16% of them lost from the follow-up, and 7% of them failed treatment [2]. In addition to the large number of missed DR-TB cases in Africa, the high proportion of unsuccessful outcomes linked with DR-TB patients will threaten the prospect of achieving the set target for the EndTB Strategy by 2035 [2, 5]. Accordingly, measuring the number of people who died from any cause during the follow-up period of standardized second-line anti-tuberculosis drugs (SLD) as the incidence of all-cause mortality and its predictors are the most informative ways of assessing the DR-TB treatment outcomes [6].

The mortality commonly occurs among the persons receiving SLD [7, 8]. Patient characteristics like older age, male sex, underweight, comorbid conditions including HIV-coinfection, and extra-pulmonary involvement are the frequent explanations to contribute to the increased incidence of mortality among the persons receiving SLD [9, 10]. Besides, a high incidence of

mortality was also reported among the DR-TB patients with previous history of TB and those patients with features involving undernutrition and excessive alcohol use [11, 12].

Since 2010, there has been a rapid evolution on the better use of more effective DR-TB treatment regimens, mainly in African and Asian patients [2]. There have also been progress in discovering novel drugs and approaches to the use of repurposed drugs, and some of the world countries have begun adding these medicines to the standardized SLD regimen [13–17]. Again, there have been advances with respect to rapid testing, detection, and effective treatment with shorter regimens for the DR-TB patients [18–22]. In line to these changes, the average annual all-cause mortality rate in resource-limited settings looked to mirror the global average, but it remained unacceptably high and reaches up to 30 percent or above for some of the resource-limited countries including sub-Saharan Africa (SSA) [2, 8, 23]. Indeed, a reduction in the mortality rate has been predicted in line with the various changes implemented in these countries as part of the EndTB Strategy target set for 2035 [2]. Accordingly, there appears to be other factors than the treatment features that could influence the all-cause mortality among persons receiving the SLD therapy. Understanding such potential factors can inform a policy priority for the SLD therapy alongside its fortifications with novel or repurposed drugs. As such, a focused evidence that considers the combined risks of behavioral, sociodemographic, and clinical features of patients with proven and consistent influences on the high incidence of mortality among persons receiving the SLD therapy is mandatory. This evidence can inform an appropriate and a context-led approach with the potential to contribute to the DR-TB treatment successes. We thus aimed to estimate a pooled incidence proportion of all-cause mortality and its predictors among the persons receiving SLD treatments in SSA.

## Methods

A methodological protocol for this review was prepared according to a statement recommendation made by the Preferred Reporting Items for Systematic Review and Meta-Analysis Protocols (PRISMA-P) in 2015 [24]. The International Prospective Register of Systematic Reviews (PROSPERO) has registered the protocol with a trial registration number of CRD42020160473. Besides, we strictly followed the PRISMA flow diagram during the process of study selection [25].

### Search strategy

We identified publications by systematic searches of PubMed/Medline, Embase, Scopus, Google Scholar, Heath Technology assessment and Cochrane Library, from February to March 15, 2020. The identified records were downloaded with an appropriate format and linked to the Endnote. The terms used for our search strategy included: second-line*, rifampicin-resistant, multidrug-resistant, tuberculosis, treatment outcome, unfavorable*, death, factor, and Africa, South of the Sahara. During the searches, we employed Boolean operators (AND, OR) and truncations as appropriate to identify and include more publications. A PubMed search strategy is added to the supporting information section as a (S1 Table).

### Eligibility criteria

We applied several inclusion and exclusion criteria that were defined a priori to the records identified. Publications eligible for inclusion reported on the incidence of mortality and/or its predictors among individual patients treated for DR-TB (i.e., rifampicin-resistant TB (RR-TB) or multidrug-resistant TB (MDR-TB)) as a primary outcome, or secondary outcome. The mortality outcomes considered were those encounters reported following SLD therapy initiation that also included interim reports [26]. Again, publications of studies conducted in Africa,

South of the Sahara and published from January 2010 to March 15, 2020, were included. We excluded abstracts with unrelated data, non–English language studies, publications without original data (reviews, correspondence, guidelines, letters, and editorials), and original articles that reported insufficient or irrelevant information. We also excluded studies that reported results from a case series or case report, qualitative data, and mixed findings from pre-extensively or extensively drug-resistant (pre-XDR or XDR) TB and MDR-TB patients that did not separately report outcomes for the MDR-TB. The studies with treatment outcomes for the patients with XDR-TB were excluded to avoid confusions in observed outcomes of the standardized SLD therapy, for complicated cases of the XDR-TB were assumed as different conditions from the other forms of DR-TB [27].

## Study selection procedure

Initially, we removed duplicates from the identified publications by the use of Endnote, version 8.2 (Thomson Reuters, Stamford, CT, USA) and manual screening. Next, two of us (BH and MS) independently appraised the titles and abstracts of the retained publications and selected relevant articles for possible inclusion in the review. Accordingly, the RR/MDR-TB studies with reported mortality outcome and/or its predicting factors were kept. With this, the RR-TB was defined as any resistance to rifampicin, in the form of mono-resistance, poly-resistance, or MDR but not XDR [27]. This definition included the MDR-TB [2] which is a resistance to both isoniazid and rifampicin [27]. We also considered studies that reported retreatment TB cases managed with SLD regimens among patients with failed treatment or defaulted. Finally, two of the authors (DE and FA) independently collected and evaluated full-text details of the remained articles for quality and eligibility assessment.

## Quality assessment

Methodological quality of the retained publications were appraised by two independent authors using the Joanna Briggs Institute's (JBI's) checklist for cohort studies [28]. A third author's appraisal score was considered in cases of disagreement between scores of the two authors. Finally, all studies that fulfilled at least 50% of the quality requirement as per the average positive score of the appraisers were considered for this review.

## Data extraction

A data abstraction format prepared in a Microsoft excel sheet was employed to extract all relevant information for the systematic review and meta-analysis. Two non-blinded investigators (DE and FA) extracted the data independently, reviewed it for discrepancies, and finally reached a consensus through discussion. The following variables were extracted: name of the first author; year of the publication; number of deaths reported during the SLD therapy; the total number of patients treated with the SLD; median duration of the standardized SLD therapy; design of the studies; settings; age category of patients (children, adults, children and adults); the WHO group of the SLD regimen used (group A, group B, group C); and details of the specific drugs used in the SLD regimen. We also extracted the number of persons exposed to the predictors of mortality, and the number died from those exposed and unexposed while receiving the SLD therapy. Additionally, predictors of mortality were extracted for studies that linked patient characteristics with this outcome. For detailed extraction of the drugs used, we abstracted the standardized SLD regimens that were fortified with a later generation fluoroquinolone, bedaquiline and linezolid as group A while the regimens that added cycloserine or terizidone and/or clofazimine were abstracted as group B. Likewise, regimens that included injectable aminoglycosides (kanamycin, capreomycin, or amikacin), ethambutol,

pyrazinamide, ethionamide, para-aminosalicylic acid, delamanid, etc. (i.e., to complete the therapy) were extracted as group C [29].

## Outcome definitions

The definition we considered for mortality was in line with the WHO outcomes definition among the patients with DR-TB [26]. Accordingly, the sum of cure and treatment completion was considered as a successful outcome while failed treatment, death, lost to follow-up and unevaluated outcomes were assumed as unsuccessful. In this respect, the death outcome from any cause during the time period of SLD therapy was considered as the all-cause mortality and this was the primary outcome of interest [26].

## Data analysis and synthesis

Statistical pooling for incidence proportion estimates was performed according to the random-effects model with generic inverse-variance methods, using Stata 15.0 (StataCorp. 2017, *Stata Statistical Software*: *Release 15*; College Station, TX: StataCorp LLC). The random-effects model of analysis was assumed since the studies identified were observational in nature and they had both clinical and methodological variabilities. The percentage rates of the mortality incidences were presented using forest plots. In this analysis, however, risk estimates for predictors were not pooled from individual studies as this approach would have not been feasible and valid given the high risk for bias [30]. To this end, we considered an approach suggested by Ross et al. and evaluated a given predictor with proven significant and independent association with the outcome of interest [31]. In determining the risk ratios, incidences of mortality among patients with HIV-coinfection and other comorbidities such as diabetes mellitus, myocardial infraction, congestive heart failure, asthma, hypertension, chronic pulmonary insufficiency, depression, epilepsy, etc. were pooled together. Again, the mortality incidence among persons diagnosed with anemia, underweight, pneumonia, pneumothorax, hemoptysis, nutritional problems, etc. were combined together as other clinical conditions. The forest plots were employed to present the pooled risk ratio of the factors associated with the incidence of mortality. The degree of heterogeneity for effect sizes among the studies was appraised using $chi^2$ ($I^2$) statistics. In line with this, subgroup analyses were carried out to explain few patient features with the potential to account for the differences in the effect sizes of the mortality incidence. Publication bias (or small-study effects) was assessed by a graphical inspection of funnel plot. Next, Egger's regression and Begg's correlation tests were performed to test the presence of publication bias. Lastly, all statistical tests were considered as significant for P-values less than 0.05.

## Results

### Study selection

There were 4,255 publications identified and eligibility appraised for this review. From among these 4,255 records we identified, 422 duplicates and 3,619 unrelated studies (i.e., 1,554 of them by screening titles and 2,065 of them by screening abstracts) were excluded. Next, 171 publications were excluded with reasons from among the 214 full-text details we appraised for quality and eligibility. Finally, 43 publications that met the priori eligibility and quality requirements for the review were included in the study (Fig 1 and S2 Table).

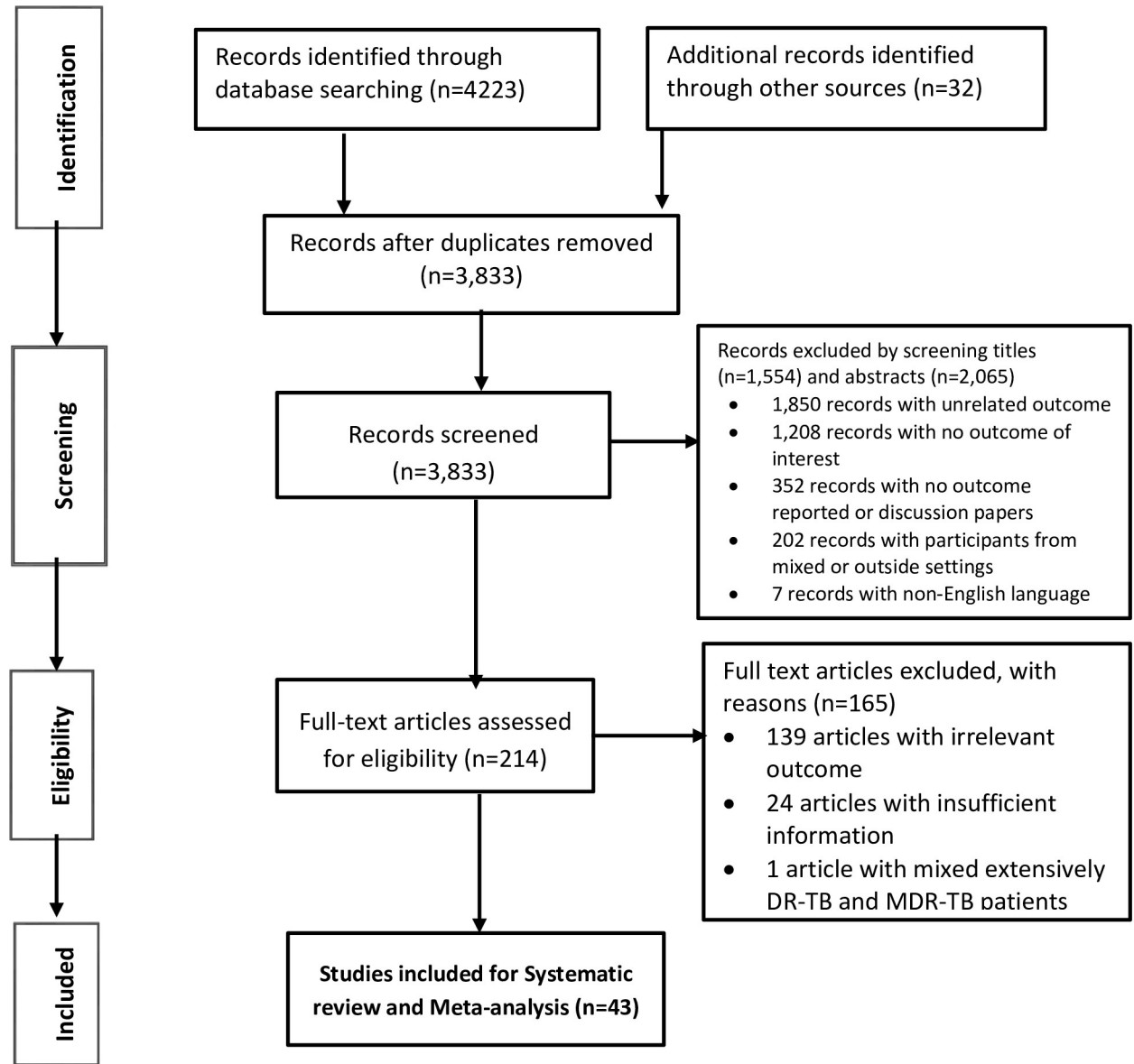

**Fig 1. PRISMA flow diagram depicting the selection process.**

## Study characteristics

From among the 43 studies that were included, a total of 31,525 persons receiving SLD were followed-up and 4,976 of them encountered the incidence of mortality. The study participants for four of these studies were children [32–35]; for 20 of the studies were adults [23, 36–54]; and for 19 of the studies were both adults and children [55–73]. Methodological design for nine of the studies was prospective cohort [23, 39, 42–46, 50, 63] while it was retrospective cohort for 34 of the remaining studies [32–38, 40, 41, 47–49, 51–62, 64–73]. In terms of regions of the SSA from where the data were originated, 25 of the studies were from the southern region [23, 32–34, 36–39, 41, 43, 45–47, 49, 51–56, 63, 66, 68, 71, 72]; 14 of the studies were from the eastern region [35, 40, 44, 48, 57, 58, 60–62, 65, 67, 69, 70, 73]; two of the studies were

from the central region [42, 64]; one of the study was from the western region [59]; and the remaining one study was from multi-sites in different SSA regions [50]. A total of 23 studies out of the 43 publications identified had reported the predictors of mortality among the persons receiving SLD therapy [23, 32, 34, 35, 37–41, 44, 47, 49, 52, 54, 55, 57, 59–61, 64, 67, 69, 71]. However, varied analytic models of analyses were used by these studies in their attempt to identify the potential predictors of mortality (Table 1 and S3 Table).

## Proportion of patients with the incidence of mortality

The pooled estimate for the incidence of mortality as a percentage rate was 17% (95% CI: 15% - 18%; $I^2$ = 91.40; P = 0.00). The effect sizes estimated for the individual studies ranged from 7% (95% CI: 3% - 16%) to 44% (95% CI: 39% - 50%) (Fig 2).

## Sensitivity analysis

To explore the source of heterogeneity among the included studies, we performed a sensitivity analysis by excluding two of the outliers [41, 42]. However, this resulted in a slight reduction to the degree of heterogeneity with a percentage decrease to the incidence of mortality that was already estimated (effect size: 16%; 95% CI: 15% - 18%; $I^2$ = 88.75%; P = 0.00) (Fig 3).

## Subgroup analyses

Since the variability among studies remained high even after the sensitivity analysis, we performed subgroup analyses to further explore the source of heterogeneity. We categorized the studies by groups of the SLD regimen, median duration of follow-up for the SLD therapy, and regions of the SSA as the key observational features. However, none of these subgroups appeared homogenous except the slight variabilities in the group-specific mortality estimates which were not statistically significant. The incidence proportion of mortality ranged from 13% (95% CI: 8%-17%) for studies with the median follow-up of less than 15 months to 18% (95% CI: 13%-19%) for studies with the median follow-up of 16–20 months. This was 13% (95% CI: 10%-17%) for group A, 15% (95% CI: 13%-17%) for group B, 19% (95% CI: 14%-24%) for group C and 21% (95% CI: 13%-29%) for non-specific SLD regimens. Again, the incidence proportion of mortality was 19% (95% CI: 17%– 22%) for studies from the southern SSA region, 13% (95% CI: 11% - 15%) for studies from the eastern SSA region, and 11% (95% CI: 6% -17%) for studies from mixed SSA regions (S1–S3 Figs).

## Predictors of mortality

A total of 23 studies had reported at least one predictor linked with the incidence of mortality among persons receiving the SLD therapy. Nineteen (19) of these studies reported HIV-coinfection and other comorbidities (i.e., diabetes mellitus, myocardial infractions, hypertension, congestive heart failure, asthma, epilepsy, depression, chronic pulmonary insufficiency and pulmonary fibrosis) [32, 34, 35, 37–41, 44, 49, 52, 54, 55, 59–61, 64, 67, 71] while ten of the studies reported diagnoses of clinical conditions such as anemia, underweight (BMI < 18.5 Kg/m2), pneumothorax, pneumonia, hemoptysis, opportunistic infections, cavitary changes and nutritional problems [23, 32, 34, 44, 47, 51, 52, 57, 64, 69] as the key predictors of all-cause mortality. Besides, three of the studies specific to each feature had reported resistance to SLD [54, 69, 71]; male sex [32, 49, 67]; and excessive substance use [61, 64, 69] as the predictors of all-cause mortality. Moreover, two of the studies reported delays (i.e., more than a month) in initiating SLD therapy [59, 61]; encounters of adverse drug events (ADEs) [40, 41], and extra-pulmonary involvement [34, 41] as the predictors of the mortality outcome. Reports of

**Table 1. Characteristics of identified publications and their analytic models of factor prediction for the SLD treatment outcomes.**

| Study | # died | Total size | Follow-up period | Design | Setting | Age category | Analytic model | Group of SLD regimen | Details of the drugs used |
|---|---|---|---|---|---|---|---|---|---|
| Adewumi (2012) | 44 | 336 | 24 months | RFU | South Africa | Adults and Children | $\chi^2$ test | Group B | kanamycin, ethionamide, ofloxacin, cycloserine, pyrazinamide |
| Alakaye (2018) | 83 | 343 | 18 months | RFU | Lesotho | Adults and Children | Cox proportional hazards regression | Group C | Amikacin, kanamycin, Capreomycin or any fluoroquinolone |
| Alene (2017) | 31 | 242 | 20 months | RFU | Ethiopia | Adults and Children | Cox proportional hazards regression | Group B | Pyrazinamide, capreomycin, levofloxacin, ethionamide, cycloserine |
| Ali (2019) | 22 | 156 | 18 months | RFU | Sudan | Adults and Children | Cox proportional hazards regression | Group B | pyrazinamide, capreomycin, levofloxacin, ethionamide and cycloserine |
| Bajehson (2019) | 38 | 147 | 20 months | RFU | Nigeria | Adults and Children | Cox proportional hazards regression | Group B | Capreomycin, levofloxacin, cycloserine, prothionamide and pyrazinamide |
| Borisov (2017) | 17 | 113 | 18 months | RFU | South Africa | Adults | $\chi^2$ test | Group A | Bedaquiline, linezolid, moxifloxacin, clofazimine and carbapenems |
| Brust (2010) | 223 | 1209 | 24 months | RFU | South Africa | Adults | Multivariate logistic regression | Group B | Kanamycin, ofloxacin, pyrazinamide, ethambutol or cycloserine and thionamide |
| Brust (2018) | 22 | 191 | 32 months | RFU | South Africa | Adults | Cox proportional hazards regression | Group B | Kanamycin, moxifloxacin ethionamide, terizidone, ethambutol and pyrazinamide |
| Cox (2014) | 128 | 718 | 24 months | RFU | South Africa | Adults/ Adolescents | Cox proportional hazards regression | Group C | Ofloxacin, kanamycin, ethambutol, ethionamide and pyrazinamide |
| Fantaw (2018) | 30 | 164 | 13 months | RFU | Ethiopia | Adults and Children | Cox proportional hazards regression | NS | Standardized SLD |
| Farley (2011) | 177 | 757 | 24 months | FU | South Africa | Adults | Cox proportional hazards regression | Group C | Pyrazinamide, ethambutol, ethionamide, ofloxacin, and either amikacin or kanamycin. |
| Getachew (2013) | 29 | 188 | 14 months | RFU | Ethiopia | Adults and Children | Cox proportional hazards regression | NS | Standardized SLD |
| Girum (2017) | 13 | 154 | 24 months | RFU | Ethiopia | Adults | Cox proportional hazards regression | Group B | Capreomycin, amikacin, ethionamide, levofloxacin and cycloserine |
| Hall (2017) | 69 | 423 | 24 months | RFU | South Africa | Children | Cox proportional hazards regression | Group C | A fluoroquilone and second-line injectables |
| Hicks (2014) | 8 | 68 | 18 months | RFU | South Africa | Children | Multivariate logistic regression | Group B | pyrazinamide, ethambutol, terizidone, kapromycin, ofloxacin and ethambutol |
| Hirasen (2018) | 37 | 240 | 12 months | FU | South Africa | Adults | Cox proportional hazards regression | NS | Standardized SLD |
| Huerga (2017) | 21 | 145 | 24 months | RFU | Kenya | Adults and Children | Multivariate logistic regression | Group B | kanamycin or capreomycin, levofloxacin, prothionamide, cycloserine, para-aminosalicylic acid |
| Jikijela (2018) | 147 | 332 | 24 months | RFU | South Africa | Adults | Multivariate logistic regression | NS | Standardized SLD |

*(Continued)*

**Table 1.** (Continued)

| Study | # died | Total size | Follow-up period | Design | Setting | Age category | Analytic model | Group of SLD regimen | Details of the drugs used |
|---|---|---|---|---|---|---|---|---|---|
| Kapata (2017) | 12 | 71 | 20 months | FU | Zambia | Adults and Children | Cox proportional hazards regression | Group B | kanamycin, levofloxacin, ethionamide, cycloserine and pyrazinamide |
| Kashongwe (2017) | 18 | 199 | 6 months | RFU | Congo | Adults and Children | Multivariate logistic regression | NS | Standardized SLD |
| Kuaban (2015) | 10 | 150 | 12 months | FU | Cameroon | Adults | Multivariate logistic regression | Group B | gatifloxacin, clofazimine, prothionamide, ethambutol and pyrazinamide |
| Leveri (2019) | 56 | 332 | 24 months | RFU | Tanzania | Adults and Children | Multivariate logistic regression | Group B | Amikacin or kanamycin, ofloxacin or levofloxacin, pyrazinamide, ethionamide, cycloserine, and ethambutol |
| Loveday (2015) | 223 | 1549 | 24 months | FU | South Africa | Adults | Cox proportional hazards regression | Group B | Kanamycin, pyrazinamide, ethambutol, ethionamide, ofloxacin and cycloserine |
| Marais (2013) | 65 | 324 | 24 months | RFU | South Africa | Adults and Children | Multivariate logistic regression | Group B | kanamycin, pyrazinamide, ofloxacin, ethionamide and terizidone or ethambutol |
| Meressa (2015) | 85 | 612 | 24 months | FU | Ethiopia | Adults | Cox proportional hazards regression | Group B | Three of levofloxacin, ethionamide, cycloserine or para-aminosalicylic acid, pyrazinamide and amikacin or kanamycin or capreomycin |
| Mibei (2016) | 18 | 205 | 20 months | RFU | Kenya | Adults and Children | Multivariate logistic regression | Group B | kanamycin, levofloxacin, cycloserine, ethionamide and pyrazinamide |
| Mohr (2015) | 123 | 757 | 18 months | RFU | South Africa | Adults and Children | Multivariate logistic regression | NS | second-line anti-TB drugs |
| Mollalign (2015) | 37 | 342 | 16 months | RFU | Ethiopia | Adults and Children | Cox proportional hazards regression | Group C | Ethambutol, streptomycin, kanamycin, amikacin and capreomycin |
| Mollel (2019) | 29 | 201 | 20 months | RFU | Tanzania | Adults and Children | Multivariate logistic regression | Group B | amikacin or kanamycin, ofloxacin, cycloserine, ethionamide, pyrazinamide and ethambutol |
| Ndjeka (2018) | 25 | 200 | 24 months | FU | South Africa | Adults | Multivariate logistic regression | Group A | Bedaquiline, clofazimine, levofloxacin, linezolid, kanamycin |
| Padayatchi (2014) | 7 | 23 | 18 months | FU | South Africa | Adults | Multivariate Poison regression | Group C | kanamycin, ofloxacin, pyrazinamide, ethambutol or cycloserine and ethionamide |
| Satti (2012) | 46 | 134 | 24 months | RFU | Lesotho | Adults | Cox proportional hazards regression | Group B | fluoroquinolone, prothionamide or ethionamide, cycloserine, pyrazinamide, para-aminosalicylic acid, etc. |
| Schnippel (2015) | 2165 | 15339 | 24 months | RFU | South Africa | Adults and Children | Multivariate Poison regression | Group B | Kanamycin or amikacin, ofloxacin, ethambutol or ethionamide, terizidone, and pyrazinamide |
| Seddon (2012) | 13 | 111 | 24 months | RFU | South Africa | Children | Multivariate logistic regression | Group B | Amikacin, capreomycin, ofloxacin, ethionamide, para-aminosalicylic acid, terizidone, linezolid, etc. |
| Shibabaw (2018) | 19 | 235 | 24 months | RFU | Ethiopia | Adults | Cox proportional hazards regression | Group B | At least three of oral agents (pyrazinamide, levofloxacin, ethionamide, protonamide, cycloserine or para-aminosalicyclic acid) and an injectable agent (amikacin, kanamycin, capreomycin) |

(*Continued*)

**Table 1.** (Continued)

| Study | # died | Total size | Follow-up period | Design | Setting | Age category | Analytic model | Group of SLD regimen | Details of the drugs used |
|---|---|---|---|---|---|---|---|---|---|
| Shin (2017) | 118 | 588 | 24 months | RFU | Botswana | Adults | Multivariate Poison regression | Group B | amikacin, levofloxacin, ethionamide, cycloserine, and pyrazinamide |
| Tola (2020) | 18 | 155 | 36 months | RFU | Ethiopia | Children | Cox proportional hazards regression | Group B | levofloxacin, ethionamide, cycloserine, para-aminosalicyclic acid, pyrazinamide, prothionamide, linezolid, clofazimine, amikacin, kanamycin and capreomycin |
| Trebucq (2018) | 78 | 1006 | 24 months | FU | 9 Africa countries | Adults | Multivariate logistic regression | Group B | moxifloxacin, clofazimine, ethambutol, pyrazinamide, kanamycin, prothionamide and high-dose isoniazid |
| Umanah (2015a) | 181 | 947 | 24 months | RFU | South Africa | Adults | Multivariate logistic regression | Group B | Kanamycin/Amikacin, Moxifloxacin, Ethionamide, Terizidone, Ethambutol and/or pyrazinamide |
| Umanah (2015b) | 258 | 1137 | 24 months | RFU | South Africa | Adults | Multivariate Poison regression | Group B | kanamycin/amikacin, moxifloxicin, ethionamide, terizidone, and pyrazinamide |
| Van der Walt (2016) | 123 | 393 | 24 months | RFU | South Africa | Adults and Children | $\chi^2$ test | NS | Standardized SLD |
| Verdecchia (2018) | 37 | 174 | 18 months | RFU | Eswatini | Adults | Cox proportional hazards regression | Group B | Levofloxacin, ethionamide, terizidone or cycloserine, pyrazinamide, kanamycin/amikacin with/without para-aminosalicylic acid |
| Woldeyohannes (2019) | 73 | 415 | 20 months | RFU | Ethiopia | Adults and Children | Cox proportional hazards regression | Group B | Pyrazinamide, Ethambutol, Capreomycin, Levofloxacin, Ehionamide, Cycloserine; and others |
| **Total** | **4,976** | **31,525** | | | | | | | |

Note

#, number; NS, not specified; and $\chi^2$, chi-squared.

individual studies also indicated multiple other drugs than SLD [41]; previous episode of TB [71]; previous history of failed treatment [23], and under five aged children [35] as the predictors of mortality.

We combined risk ratio estimates for at least three of the studies that reported similar predictors of all-cause mortality and found its significant increases among persons receiving the SLD and with certain characteristics. These features included diagnoses of newer clinical conditions (RR: 2.36; 95% CI: 2.82–3. 05); substance use (RR: 2.56; 95% CI: 1.78–3.67); presence of HIV-coinfection and other comorbidities (RR: 1.96; 95% CI: 1. 65–2.32); resistance to SLD (RR: 1.75; 95% CI: 1.37–2.23); and male sex (RR: 1.82; 95% CI: 1.35–2.44) (Fig 4). The resistance to SLD regimen was defined by the studies as any resistance to fluoroquinolones or at least one injectable aminoglycoside (i.e., capreomycin, kanamycin, amikacin) [54, 69, 71].

## Publication bias

Graphical visualization of the funnel plot found its symmetrical appearance which gave us hint about absence of the publication bias. This visual inspection was further tested by using Egger's regression and it also showed no evidence of the small-study effects (effect estimate: 1.40; 95% CI: -0.14–2.94; P = 0.073). Additionally, Begg's correlation test revealed no evidence of the publication bias (Z = 0.96; P = 0.34) (Fig 5).

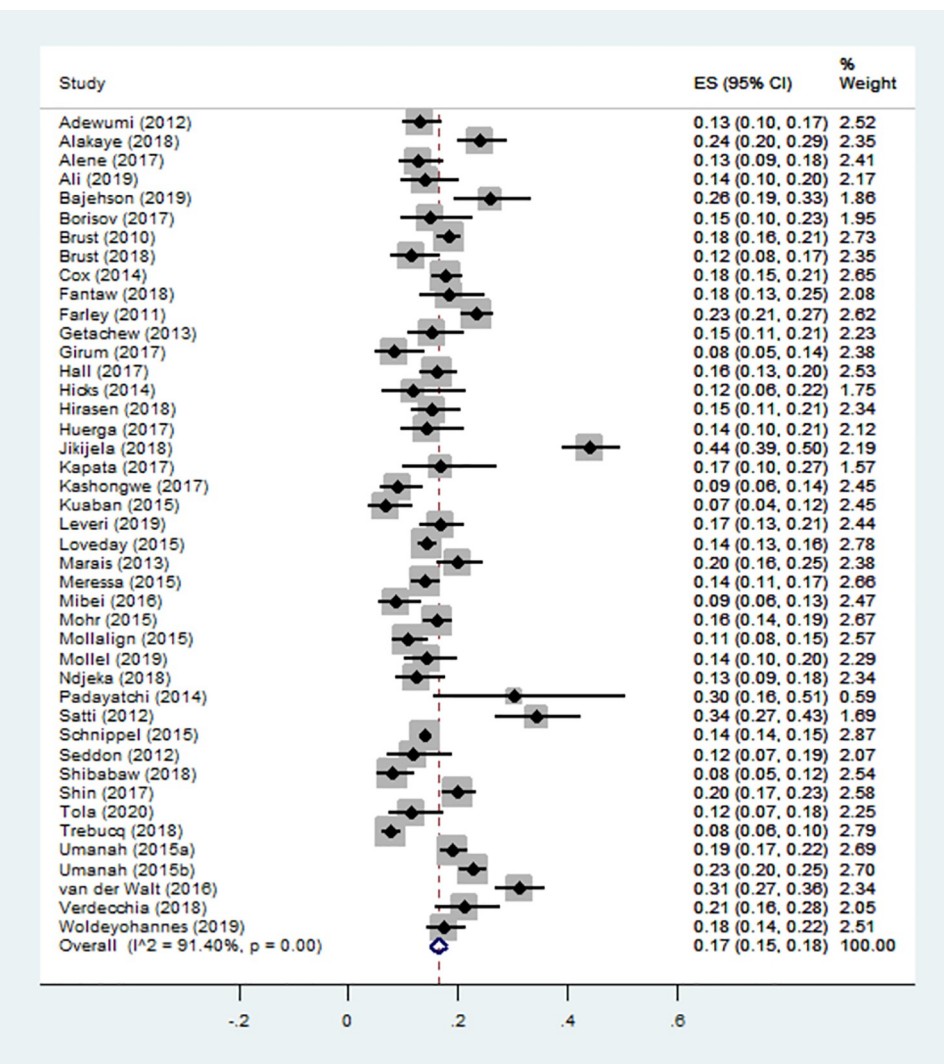

**Fig 2. Forest plot for the incidence proportion of mortality.**

## Discussion

More than one-sixths of the persons receiving SLD for DR-TB managements in SSA had died during the last decade. A relatively lower incidence proportion of the mortality was found among the DR-TB patients treated with group A or B regimens compared with those patients treated with group C regimens. The pooled risk ratio estimate for the identified predictors found increased incidences of mortality among persons with features of established comorbidities, diagnoses of newer clinical conditions, resistance to SLD, substance use, and male sex.

The pooled incidence proportion of mortality among the DR-TB patients treated with SLD in SSA was 17%. No significant differences were found by subgroup analyses that considered the median time period of treatment follow-up, specific regimen groups for SLD and regions of the SSA. However, the pooled estimate of mortality was as high as 19% in southern SSA and as low as 13% among the patients treated with group A SLD. It was also estimated to be 15% among the patients treated with group B regimens. Essentially, these estimates are promising and can hint as though this setting is on a right track to achieving the EndTB strategy target of

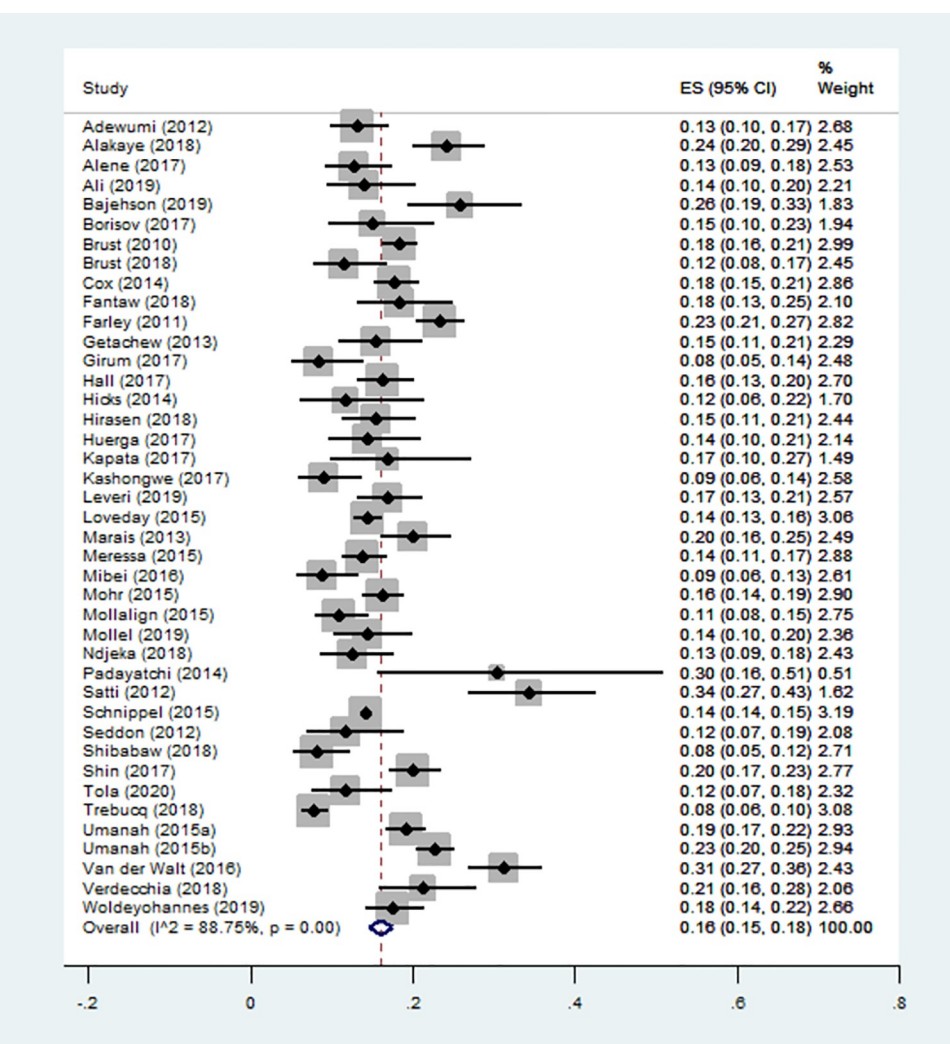

**Fig 3. Forest plot for incidence proportion of mortality by excluding outliers.**

90% mortality reduction by 2030 [2]. A study report also revealed similar benefits of using SLD regimens that contained linezolid, a later generation fluoroquinolones (levofloxacin or moxifloxacin), bedaquiline, clofazimine, or capreomycin [74]. Most of these drugs are the essential medicines that are usually added to groups A, B, or shorter SLD regimens. A similar meta-analysis also reported that there have been no significant differences in survival benefits with respect to specific drugs used and the time period of the treatment follow-up [75]. Again, most of the specific drugs added to group A, B or C regimens were part of the DR-TB treatment advances in the last decade [76]. Similar benefits were reported by the use of repurposed drugs, newer drugs, later generation fluoroquinolones, ethionamide or prothionamide, four or more effective drugs in intensive phase, and three or more likely effective drugs in continuation phase of the DR-TB treatment [76]. Most of the DR-TB patients included in this review also used SLD regimens that added the later generation fluoroquinolones, repurposed agents, newer drugs, and injectable aminoglycosides. Besides, these drugs had appeared to have strong association with treatment success and survival benefits [77].

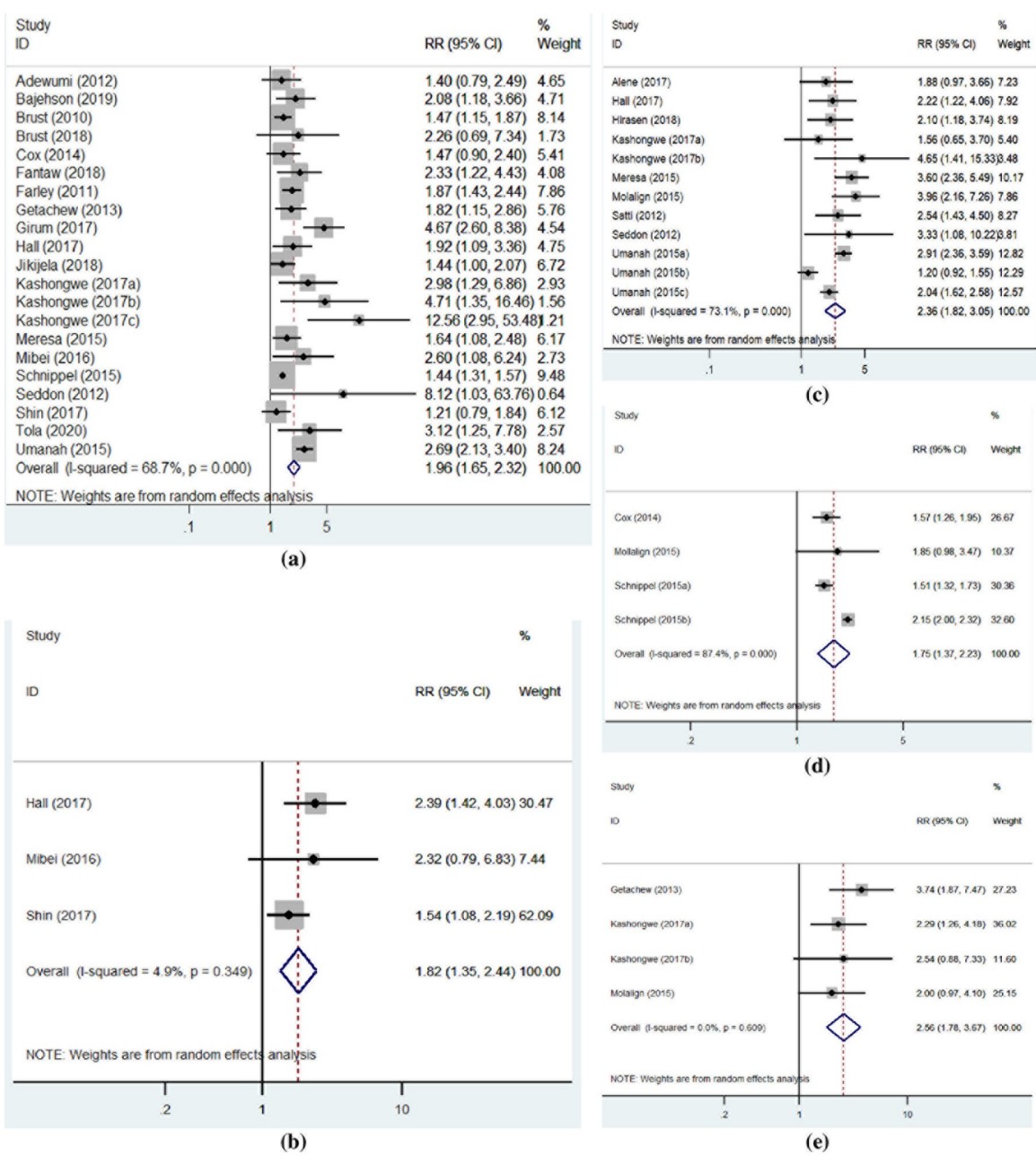

**Fig 4. Predictors of mortality among persons receiving SLD in SSA.** (a) Forest plot describing presence of HIV and other comorbidities as the predictors of mortality. (b) Forest plot describing male sex as a predictor of mortality. (c) Forest plot describing diagnoses of clinical conditions as the predictors of mortality. (d) Forest plot describing resistance to SLD as a predictor of mortality. (e) Forest plot describing substance uses as the predictors of mortality.

Moreover, most of the DR-TB patients in SSA had already initiated newer agents and shorter SLD regimens as part of the response to EndTB Strategy and WHO recommendation [76]. These shorter regimens appeared to be effective in treating MDR-TB [78]. With this, the type of drug added to the SLD regimen is considerable for optimal benefits [79]. To this end, the global average mortality of 15% among the cohort of DR-TB patients treated with SLD mirrors the percentage rate this study found (17%) for a similar setting (i.e., SSA) [2]. About

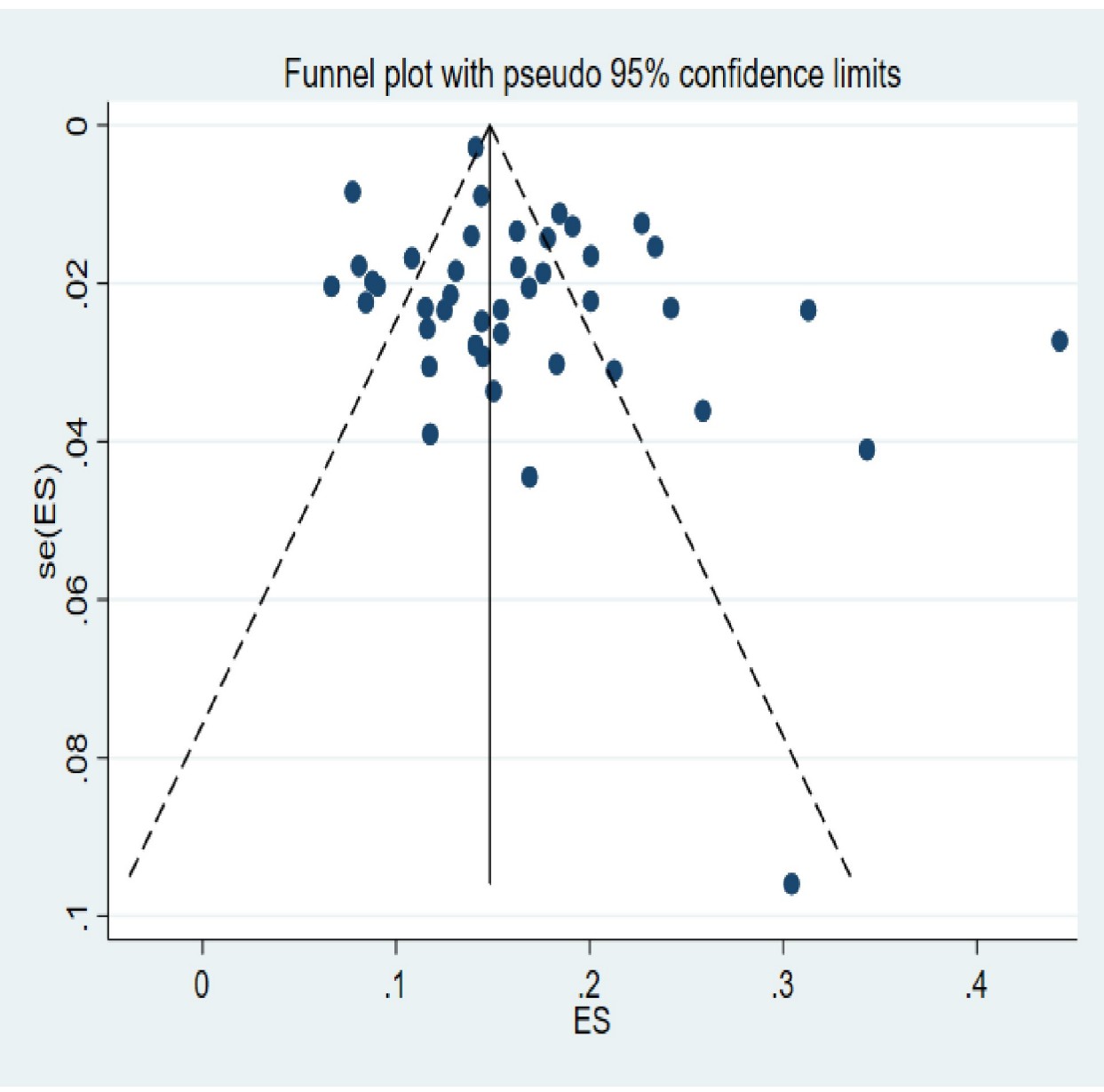

**Fig 5. Funnel plot of standard error by effect size for publication bias.**

half of countries (n = 21) from among 48 countries with high burden of DR-TB patients included in the cohort were from SSA, for which the estimated treatment successes ranged from zero percent (in Angola) to 88% (in Congo). For this cohort, majorities of the SSA countries achieved a success rate of around 65% [2]. Individual studies that participated DR-TB patients during their SLD therapy follow-up in India, Malaysia, and Pakistan also found 16%, 15.3%, and 17.4% mortality rates, respectively [80–82].

HIV and other comorbidities were the most common predictors of mortality which were frequently reported among persons receiving SLD therapy. The mortality was 1.96 times higher among the patients with comorbidities than those patients free of these conditions. Similarly, in reference to patients free of comorbidities, the risk of mortality among the

patients with various comorbid conditions while on SLD therapy was variably increased by 1.96 (95% CI: 1.35–3.85), 2.33 (95% CI: 1.34–4.05), 2.6 (95% CI: 1.82–3.70), 5.42 (95% CI: 2.66–11.04), and 6.82 (95% CI: 2.16–21.50) folds [9, 12, 80, 83]. Again, several other studies indicated a varied but increased likelihood of mortality by 1.46 (95% CI: 1.05–1.96), 1.50 (95% CI: 1.20–1.90), 1.70 (95% CI: 1.20–3.10), 1.89 (95% CI: 1.02–3.52), 2.97 (95% CI: 1.41–6.24), 3.18 (95% CI: 1.05–9.69), 3.47 (95% CI: 1.02–11.64), 4.22 (95% CI: 2.65–6.72), and 5.6 (95% CI: 3.2–9.7) times higher among the HIV-infected patients than HIV-uninfected ones [83–91]. A meta-analysis also reported 1·8 (95% CI: 1·5–2·2) times higher odds of mortality among the HIV-positive patients on antiretroviral therapy (ART) than those who did not receive the ART [92], but majorities of the studies included in our review did not report mortality by the ART status and we were unable to explore this difference by the ART treatment. Another meta-analysis also stressed on a closer link between HIV-infection and MDR-TB and found that the MDR-TB was 2.28 times more likely in HIV-infected people than those people who were HIV-uninfected [93]. In the current review, the risk of mortality was somewhat lower than the likely risks reported by several of the previous studies. Unarguably, the MDR-TB management approach of countries (in the last decade) had involved early and rapid diagnosis with geno-type testing; prompt treatment with appropriate regimens based on drug-susceptibility testing; preference for shorter regimens by using newer or repurposed drugs; a patient-centered approach; and strong infection-control measures. All these strategies might have helped the mortality reduction among the DR-TB patients with comorbidities including HIV-coinfection [94]. Also, the use of a differentiated care approach, the demand created for effective TB/HIV service delivery, the establishment of HIV/TB coordination mechanisms, the rapid scale-up of facilities with decentralization of treatment services, the regular joint supervision, and moni-toring might have contributed to the successes [95, 96].

In another way round, CD4 values lower than 50 cells/mm$^3$ (HR, 4.64; P = 0.01) and 51–200 cells/mm$^3$ (HR, 4.17; P = 0.008) among the treated DR-TB patients were found as the inde-pendent risk factors of mortality compared with those patients with CD4 values >200 cells/mm$^3$ [97]. A higher susceptibility to opportunistic infections (OIs) due to the lower immune status (indicated by the low CD4 levels) can justify this finding. Previous OIs among patients treated with SLD therapy were also related to a 3.13 (95% CI: 1.64–5.96) times higher hazards of mortality than those without the OIs episode [90]. Again, patients with previous TB history and treated with SLD for about 2–6 months had 1.46 times higher risks of mortality compared to the patients without previous TB episodes [98]. In addition, history of previous TB increased the risk of death by 1.61 fold among patients with the episode compared to those patients free of the episode [12]. A study finding also implicated numbers of the previous TB episodes to have direct links with the increased risks of death [83].

Resistance to SLD was another predictor of mortality with about 75% increased risks of death among the patients who experienced drug resistance. Consistent with this finding, a study in Brazil found that MDR-TB patients who developed resistance to SLD had 74% higher risks of death than those patients who did not experience resistance [84]. Another study also reported resistance to the SLD as a key predictor of poor outcome (OR: 2.61; 95% CI: 1.61–4.21) [81]. Besides, a 31.4% incidence proportion of mortality was reported among the cohort of patients with any form of resistance to SLD therapy [99]. This mortality could reach more than 50% among under-treated patients with the resistant strains [100]. Again, MDR-TB patients with any form of resistance to SLD were found to have the lowest success rate (29.3%) [101]. In line with this, delay in initiating the SLD regimens or substituting the regimen com-ponents with resistance could be the likely reason for extending to a further resistance. About two-fold increased odds of dying was reported with the delay in starting SLD treatment [102]. Again, near to 30% increased risk of unfavorable outcome was explained with more than a

month delay in initiation of SLD after the resistance detection [103]. Delay in the resistance detection was also reported to increase the probability of mortality by 8.3% among the patients treated with SLD therapy [104].

In this review, the risk of mortality was 2.36-fold increased among the cohorts of patients diagnosed with clinical conditions than those patients free of the conditions. Underweight and anemia were the most frequent diagnosis that the studies reported. Similarly, other studies also revealed underweight to be 1.30 (95% CI: 1.0–1.50), 2.50 (95% CI: 2.10–2.90), 2.50 (95% CI: 1.70–3.5), and 3.39 (95% CI: 1.20–9.45) times more likely related with unfavorable outcomes than the patients with normal body weight [11, 91, 105, 106]. Besides, there were pieces of evidence that reported findings of baseline underweight among most MDR-TB patients (i.e., up to 86.6%), in which anemia was the most common clinical condition (i.e., up to 73.83%) [107–109]. Besides, underweight patients had a 90% increased incidence of mortality than the patients with normal body weight [110]. In fact, severe anemia and malnutrition were known as the independent predictors of early mortality in TB patients [111].

The DR-TB patients who used excessive substances (cigarette and alcohol) and male patients had 2.56 and 1.82 times higher risks of mortality than the patients who did not use the substances and who were females, respectively. Consistently, patients with excessive substance uses tended to have poor MDR-TB treatment outcomes [112]. In a study report, patients with alcohol misuse had a 1.45 (95% CI: 1.21–1.75) times higher risks of unsuccessful outcomes than the patients who did not drink alcohol [113]. Again, MDR-TB patients with habits of cigarette smoking were found to have 5.44 (95% CI: 1.09–27.19) times higher odds of mortality than those patients who were non-smokers [89]. In line with this, a two-fold increased odds of substance abuse disorders were correlated with male sex [114]. And, the risk of mortality was 1.4 (95% CI: 1.1–1.7) and 2.0 (95% CI: 1.27–3.14) times higher among males than females on SLD therapy follow-up [9, 106].

Despite the large size of aggregate data pooled together for summary effects in this systematic review and meta-analysis, it is not without limitations. First, the studies considered for the meta-analysis were observational by nature. This selection might have resulted in a higher degree of heterogeneity with a range of potential biases. However, we employed a random-effects model to account for the anticipated heterogeneity. Second, there were some inconsistencies in the studies included in terms of the median time period of follow-up for the SLD therapy, but we assumed the intention-to-treat approach and considered deaths reported at any time during the follow-up period. Third, the retrospective cohort studies included in our review did not adjust for mortality in patients with lost follow-up and failed treatment. Due to the aggregate data meta-analysis approach and a limited control over the data, these deaths could not be accounted for, and this gap might have under-estimated the incidence of all-cause mortality. Fourth, we included articles written in the English language, and this restriction could have under-or over-estimated the pooled incidence of mortality and its predictors while on the SLD therapy. Fifth, there appeared some overlaps of included patients in four of the South Africa studies reporting national treatment register and community-based programs, but we were unable to avoid such overlaps for we do not have clear knowledge of the data source. Therefore, interpretations for the findings in this review need to be aligned and seen in contexts of these limitations.

## Conclusions

We found about one in six persons who received SLD in SSA had died in the last decade. This pooled incidence proportion of mortality while on the SLD therapy follow-up among the patients in SSA mirrors the global average mortality. Several measures including fortification

of newer or repurposed drugs and the initiation of shorter regimens were among the essential components for this acceptable mortality rate we estimated which was in line with the set target of EndTB Strategy. Nonetheless, the incidence of mortality was considerably high among DR-TB patients with comorbidities; diagnoses of other clinical conditions; resistance to SLD therapy; male sex; and excessive substance use. Therefore, modified measures involving shorter SLD regimens fortified with newer or repurposed drugs, differentiated care approaches, and support of substance use rehabilitation programs can help improve the treatment outcome of persons with the drug-resistant tuberculosis.

## Supporting information

**S1 Fig. Forest plot of mortality proportion by median duration of SLD therapy.**
(TIF)

**S2 Fig. Forest plot of mortality proportion by group of SLD regimen.**
(TIF)

**S3 Fig. Forest plot of mortality proportion by SSA regions.**
(TIF)

**S1 Table. PubMed search strategy and results.**
(DOCX)

**S2 Table. Quality assessment for the included studies.**
(DOCX)

**S3 Table. Completed PRISMA checklist.**
(DOC)

## Acknowledgments

We extend our acknowledgment to Tara Wilfong (Ph.D.) for her editorial supports in this manuscript preparation.

## Author Contributions

**Conceptualization:** Dumessa Edessa.

**Data curation:** Dumessa Edessa, Fuad Adem, Bisrat Hagos, Mekonnen Sisay.

**Formal analysis:** Dumessa Edessa, Fuad Adem, Bisrat Hagos, Mekonnen Sisay.

**Investigation:** Dumessa Edessa, Fuad Adem, Bisrat Hagos, Mekonnen Sisay.

**Methodology:** Dumessa Edessa, Fuad Adem, Bisrat Hagos, Mekonnen Sisay.

**Software:** Dumessa Edessa.

**Supervision:** Dumessa Edessa, Fuad Adem, Bisrat Hagos, Mekonnen Sisay.

**Validation:** Dumessa Edessa, Fuad Adem, Bisrat Hagos, Mekonnen Sisay.

**Writing – original draft:** Dumessa Edessa.

**Writing – review & editing:** Dumessa Edessa, Fuad Adem, Bisrat Hagos, Mekonnen Sisay.

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
