## [Decision Letter · Decision Letter 0]

2 Mar 2021

PONE-D-20-25643

Risk factors of mortality during course of second-line tuberculosis treatment in sub-Saharan Africa: A meta-analysis of cohort studies

PLOS ONE

Dear Dr. Edessa,

Thank you for submitting your manuscript to PLOS ONE. After careful consideration, we feel that it has merit but does not fully meet PLOS ONE’s publication criteria as it currently stands. Therefore, we invite you to submit a revised version of the manuscript that addresses the points raised during the review process.

In addition to the questions and concerns raised by the Reviewers, the manuscript requires professional language and grammatical editing. 

We look forward to receiving your revised manuscript.

Kind regards,

Denise Evans, PhD

Academic Editor

PLOS ONE

Journal Requirements:

2.We suggest you thoroughly copyedit your manuscript for language usage, spelling, and grammar. If you do not know anyone who can help you do this, you may wish to consider employing a professional scientific editing service.  

3. Please confirm that you have included all items recommended in the PRISMA checklist including the full electronic boolean search strategy used to identify studies with all search terms and limits for at least one database. Please attach this as supplementary file

4. In your methods section, please provide the dates when you accessed the data used for your study (it appears to be February to March 2020), as well as the publication date range that was used to search for the manuscripts included in your study.

Reviewers' comments:

Reviewer's Responses to Questions

**Comments to the Author**

1. Is the manuscript technically sound, and do the data support the conclusions?

Reviewer #1: Partly

Reviewer #2: Yes

2. Has the statistical analysis been performed appropriately and rigorously? 

Reviewer #1: Yes

Reviewer #2: I Don't Know

3. Have the authors made all data underlying the findings in their manuscript fully available?

Reviewer #1: No

Reviewer #2: Yes

4. Is the manuscript presented in an intelligible fashion and written in standard English?

Reviewer #1: No

Reviewer #2: Yes

5. Review Comments to the Author

Reviewer #1: Section of the paper Comment

General

Content I think that the authors need to highlight what makes this study novel as there have been similar systematic reviews and meta-analyses investigating factors associated with ones DR-TB prognosis, including mortaltiy.

Methodology I generally think that the methodology used requires improvement. The inclusion criteria are not explicit and I think there is a possibility that patients have been included more than once in the analysis. Additionally, I think that the intent to treat methodology might have led to an underestimation of mortality. With that said I wonder how the risk can be accurately determined.

Editing The authors should pay attention to the grammar used throughout. The grammar used needs improvement. The authors might benefit from an editing service to improve the structure and flow of many of these sections.

The authors should ensure that they define all abbreviated terms upon first time use and thereafter they can use the abbreviated term (i.e. SLD).

The authors should spell out numbers less than 10 unless they are linked to a unit of measurement.

Title

Language and representativeness The English used in the title is not clear. You might want to consider an alternative title for example:

Risk factors of mortality among persons receiving second-line tuberculosis treatment in sub-Saharan Africa: A meta-analysis of cohort studies

Additionally, in the title you might want to consider including the number of studies included in the meta-analysis.

Abstract

Introduction I think that the authors need to consider highlighting what makes this study novel here.

Methods I think that the authors need to further expand on their inclusion and exclusion criteria here.

Background

General I recommend that you update the references to include the data from the WHO Global TB report from 2020.

Kindly clarify the language used after reference 6. I think you are referring to the missing TB cases. You could say something like in addition to the large number of missing incident DR-TB cases,….”

I recommend that you edit the first sentence of the second paragraph. The wording is quite awkward. In fact I recommend that this whole paragraph is edited in order to improve the flow and use of English language.

In the last paragraph, I think you need to focus more on what makes your study unique and novel as there have been various systematic reviews which have investigated the various DR-TB treatment correlates associated with outcomes.

Methods

The protocol and registration This paragraph requires editing for improved English language use.

Search strategy I wonder why the authors did not use the term mortality in their search strategy?

Eligibility criteria I am not sure what you mean by this? Do you mean you excluded studies that only evaluated treatment success? All DR-TB outcomes are intertwined. Even patients who are LTFU or whom fail treatment are at increased risk of mortality.

“But, we excluded records with outcomes unrelated or irrelevant to mortality and/or its risk factors.”

What do you define as insufficient as mentioned in the below comment? Additionally I am not sure as to your intended meaning when you say mixed patients? I am not sure how patients were excluded on these bases. This requires clarification to truly understand who was vs was not enrolled.

“Again, we excluded studies which report outcomes with insufficient quality and no separate result in the case of mixed patients from both extensively drug-resistant TB and multidrug-resistant TB treatments.”

I wonder if the authors only included studies which used the WHO definitions of outcomes.

Again, were only 24 month outcomes evaluated? Or were interim outcomes evaluated in this study.

Study selection procedure If patients were retreatment cases, how did you decide which treatment episode to include in your study?

Data extraction process As you say that you used intention to treat analysis, does this mean you included studies that reported on interim outcomes. Again, this requires clarification in the eligibility criteria section of the paper.

Outcomes definition

Page 4, line 36

Page 4, line 39

Rather say “Outcome definitions”

The primary outcome in this study was the occurrence of AEs.

Results

Study characteristics I recommend that you do not say mixed-age patients. You could consider something like included both “adults and children”.

Why do you say 9 of the 34 included studies but then you also say that only 43 studies were included.

Table 1 I see that you include patients with different follow-up durations i.e. you did not evaluate end of treatment outcomes and you including interim treatment outcomes. These studies might therefore underreport mortality. How did you account for this in your analysis?

There could be the same patients from the study conducted by Schnippel et al 2015 in the other South Africa based cohort studies as the study conducted by Schnippel reviewed the national treatment register. Have you ensured that you excluded duplicate records for all of these studies.

Risk Factors of Mortality I see that the authors report on substance use as a risk factor for mortality but it would be helpful to have a better understanding of how this was quantified/classified in the included studies.

You say “experience of treatment failure” was a risk factor for mortality so did you evaluate all cause mortality in this study (i.e. mortality that also occurred after LTFU or Rx failure). This requires clarification in the methods section of the paper.

Later in this section you state the following:

“However, the risk of mortality was not significantly increased in under-five children, in patients who failed treatment and those patients who experienced ADE during course of the treatment.”

This seems to contradict the statement I mention in my above comment regarding treatment failure.

In your analysis of risk of mortality, did you look at all into the regimens received/the drugs included in the regimen and what drugs might have influenced mortality. Did you look at the cohort based on year of Rx initiation? Various factors have changed regarding policy and clinical practice from 2010-current and therefore I wonder how the authors accounted for the change in the management of the disease over time.

I wonder if there were any social factors you could assess as well in order to determine risk factors associated with mortality. i.e. household income, female led households, number of people in the household, setting/housing structure

Discussion

First paragraph I recommend that you first summarize your finding without yet making reference to other studies in detail. Additionally, I would not say that the mortality is justified because then it sounds as if we are saying that it is “OK” that the mortality for these individuals is acceptable. We might be able to explain the factors contributing to it but it is far from acceptable and no where in line with making progress towards achieving the goals outlined in the EndTB strategy.

You could just generally compare your mortality rate to that reported by the WHO for all patients enrolled on treatment globally as a baseline and then delve more specifically into country level mortality.

General I think this section requires many improvements. It seems that you just compare your findings to those from previously published studies rather than really looking into the policy and implementation factors that might be driving these risk factors. You might want to consider mentioning different models of care or approaches to TB control and management that can be employed to mitigate these risk factors, which have already been well documented in the literature. I do not think that the risk factors you point to here are different in any way from those already known and well documented in the literature.

Limitations

General comments I do not think that the intention to treat approach is realistic to procure good enough evidence in this study to inform policy regarding mortality. Without systematic follow-up for everyone mortality could be underestimated and the identified risk could therefore be less than that in reality should we have complete follow-up for this cohort of patients.

Conclusions

General comments I feel as though this section is weak. Although you have summarized your key findings and made recommendations they are very general. I think you could have better utilized the discussion section to expand on the various measures that could be employed to address high rates of mortality.

Figures

General comment Ensures that your figure quality is improved as the figures are blurry.

Reviewer #2: In this meta-analysis the authors set out to identify risk factors for mortality during treatment for RR/MDR-TB in sub-Saharan Africa. The central findings of the analysis include the following: 17% of patients died during RR/MDR-TB treatment, and risk factors included low BMI (RR 2.5), substance use (or abuse?) (RR 2.5), HIV (RR 2.2), co-morbid conditions (RR 2.0), resistance to SLD (RR 1.92), anemia (RR 1.75), male sex (RR1.8) and delayed initiation of RR/MDR treatment for > 1 month (RR1.6). The methodology appears scientifically sound however this manuscript requires professional language and grammatical editing prior to publication

Introduction

The introduction cites outdated references and incorrect statistics.

- (page 3) “As a result, only near to half of the drug-resistant TB patients are successfully treated; with approximately up to 40% of them dying during the treatment” – this statement is inaccurate. According to the WHO 2020 Global TB report, for the 2017 global cohort, 57% of RR/MDR-TB patients completed treatment successfully, 15% died, 16% were lost to follow-up and 7% failed. These figures have remained fairly stable for several years (see WHO Global TB report 2020 page 107).

- (page 3) “It (mortality) was alarmingly increased to over 30-40 percent in resource-limited settings”. This is not an accurate reflection of the WHO data on RR/MDR-TB mortality in sub-Saharan Africa where treatment success rates and mortality mirror the global averages. If the authors are including in this statement people who die without accessing treatment (undiagnosed or untreated) then they should clarify this sentence.

- Reference #2 and 5: Should cite updated statistics from the WHO 2020 global health report (introduction).

Methods:

- What search terms were used? The authors should strongly consider including a table describing the search terms used in the study in a supplementary appendix

- Was this an individual patient meta-analysis or aggregate data meta-analysis? This should be stated.

- Restrictions on studies: the authors only mention dates and English language as restrictions, however based on the final selection there must have been other restrictions. Possible restrictions that the authors don’t mention but that would have factored into their selection include:

o Exclusion of clinical trials

o Limiting the studies to patients with rifampicin-resistant and/or multi-drug-resistant TB

o Were reports of pre-XDR/XDR of FQ-R TB excluded or included?

- How many investigators extracted the data? Was it done concurrently or separately and were differences in data extraction reviewed?

- What data was extracted from the reviewed studies

- How was substance abuse/use defined in the studies that evaluated this as a risk factor for mortality?

- How was second-line drug-resistance defined (i.e. resistance to aminoglycosides and/or FQs) in the studies that included this data? And what were the proportions of patients with AG vs FQ resistance in those included in the review.

- The authors should define rifampicin resistant and multi-drug resistant TB in the methods section

- The failure to include ART as a variable in the risk factor analysis for patients with HIV is problematic. If possibly I would strongly suggest that the analysis be reviewed with this risk factor as it has major implications for mortality of patients with RR/MDR-TB and HIV. ART status should be defined as on ART, not on ART or ART status not known.

- Definition of treatment outcomes should be included (i.e. how were success, lost to follow-up and failure defined in the included studies).

Results

- Table 2: Insufficient detail provided on the standard treatment regimens used in the various studies. There has been rapid evolution since 2010 in the treatment of RR/MDR-TB and modifications in the treatment regimen could result in some of the heterogeneity in the results. Consider adding this data to table 2 (for example a column with % of patients treated with AG, % of patients treated with later-generation FQ and % of patients treated with BDQ for each study, if this data is available). If this level of detail is not available or not possible to retrieve, a narrative description of the regimens used in the included studies should be added to the methods section. They could be classified according to the WHO effectiveness categories in use at the time of the study (prior to the recent major change in WHO classification of second-line TB drugs).

- Table 2: should include % HIV prevalence and % on ART in each study. The study period for each included study should also be added,

- The proportion of patients with HIV on ART is not addressed in the results section and this has major implications for mortality.

Discussion

The discussion section needs to be re-written. The purpose of the discussion section is to highlight the findings of the study and place them in the context of the literature in the field. This was not done successfully. Instead the current discussion is a re-iteration of the results of the study without much additional insight or contextualisation. Many prominent recent articles in the field and important concepts are not highlighted.

The failure to discuss the impact ART on RR/MDR-TB treatment outcomes in people living with HIV stood out. Much has been written about this in the literature that is not cited or discussed.

A brief literature review revealed several relevant and important articles in this field that are not discussed or referenced suggesting that the authors have not performed an adequate literature review which may be why they appear to have difficulty contextualising their findings in the discussion.

- Bastos et al ERS 2017 - An updated systematic review and meta-analysis for treatment of multidrug-resistant tuberculosis

- Bisson et al Lancet 2020 - Mortality in adults with multidrug-resistant tuberculosis and HIV by antiretroviral therapy and tuberculosis drug use: an individual patient data meta-analysis

- Mesfin et al Plos One 2014 - Association between HIV/AIDS and Multi-Drug Resistance Tuberculosis: A Systematic Review and Meta-Analysis

- Khan et al. ERS 2017 Effectiveness and safety of standardised shorter regimens for multidrug-resistant tuberculosis: individual patient data and aggregate data meta-analyses

- Ahmad et al. The Lancet 2018. Treatment correlates of successful outcomes in pulmonary multidrug-resistant tuberculosis: an individual patient data meta-analysis

- Isaakidis et al IJTLD 2015. Treatment outcomes for HIV and MDR-TB co-infected adults and children: systematic review and meta-analysis.

- Fox et al. ERS 2017 Group 5 drugs for multidrug-resistant tuberculosis: individual patient data meta-analysis

6. PLOS authors have the option to publish the peer review history of their article (what does this mean?). If published, this will include your full peer review and any attached files.

Reviewer #1: No

Reviewer #2: **Yes: **Rebecca H Berhanu

---

## [Author Response · Author response to Decision Letter 0]

19 Apr 2021

Authors' response to reviewers comments are attached a separate file with all files submitted.

---

## [Decision Letter · Decision Letter 1]

8 Oct 2021

PONE-D-20-25643R1Mortality and its risk factors among persons receiving second-line tuberculosis treatment in sub-Saharan Africa: A meta-analysis of 43 cohort studiesPLOS ONE

Dear Dr. Edessa,

Thank you for submitting your revised manuscript to PLOS ONE. After careful consideration, we feel that it has improved but still does not fully meet PLOS ONE’s publication criteria as it currently stands. Therefore, we invite you to submit a revised version of the manuscript that addresses the points raised during the review process. In particular Reviewer #1 still has minor concerns; in addition your manuscript was reviewed by a professional statistician, who raised several concerns that should be addressed before we can accept you manuscript for publication.

We look forward to receiving your revised manuscript.

Kind regards,

Olivier Neyrolles

Academic Editor

PLOS ONE

Journal Requirements:

Reviewers' comments:

Reviewer's Responses to Questions

**Comments to the Author**

1. If the authors have adequately addressed your comments raised in a previous round of review and you feel that this manuscript is now acceptable for publication, you may indicate that here to bypass the “Comments to the Author” section, enter your conflict of interest statement in the “Confidential to Editor” section, and submit your "Accept" recommendation.

Reviewer #1: (No Response)

Reviewer #3: (No Response)

2. Is the manuscript technically sound, and do the data support the conclusions?

Reviewer #1: Partly

Reviewer #3: Yes

3. Has the statistical analysis been performed appropriately and rigorously? 

Reviewer #1: Yes

Reviewer #3: Yes

4. Have the authors made all data underlying the findings in their manuscript fully available?

Reviewer #1: Yes

Reviewer #3: Yes

5. Is the manuscript presented in an intelligible fashion and written in standard English?

Reviewer #1: No

Reviewer #3: Yes

6. Review Comments to the Author

Reviewer #1: Kindly review all of my comments in the attached file. Generally, I feel as though this manuscript still requires heavy editing for improved English language and clarity.

Reviewer #3: The present is an interesting paper.

Some issues.

Abstrct should be shortened : e.g. introduction should be more focused on resistance to tubercolosis.

Abstract>primary end point should be defined

Abstract; it should be better specified how (And if ) RRs were pooled

Abstract: some incidence or percentages should be added

Methods: due to observational design of the studies probably random effect should be used

MEthods/results: comparison between different lenght of therapy is not inferental, but just observational. this should be clearly stated

Methods: authors should evalaute if they need to pool together RR (see PMID: 22360945)

7. PLOS authors have the option to publish the peer review history of their article (what does this mean?). If published, this will include your full peer review and any attached files.

Reviewer #1: No

Reviewer #3: **Yes: **Fabrizio D'Ascenzo

---

## [Author Response · Author response to Decision Letter 1]

15 Nov 2021

A document of rebuttal letter as 'response to reviewers' comments was attached alongside the manuscript files. Please check the attachment for this section as well.

---

## [Decision Letter · Decision Letter 2]

29 Nov 2021

Incidence and predictors of mortality among persons receiving second-line tuberculosis treatment in sub-Saharan Africa: A meta-analysis of 43 cohort studies

PONE-D-20-25643R2

Dear Dr. Edessa,

We’re pleased to inform you that your manuscript has been judged scientifically suitable for publication and will be formally accepted for publication once it meets all outstanding technical requirements.

Kind regards,

Olivier Neyrolles

Section Editor

PLOS ONE

Reviewers' comments:

Reviewer's Responses to Questions

**Comments to the Author**

1. If the authors have adequately addressed your comments raised in a previous round of review and you feel that this manuscript is now acceptable for publication, you may indicate that here to bypass the “Comments to the Author” section, enter your conflict of interest statement in the “Confidential to Editor” section, and submit your "Accept" recommendation.

Reviewer #3: All comments have been addressed

2. Is the manuscript technically sound, and do the data support the conclusions?

Reviewer #3: (No Response)

3. Has the statistical analysis been performed appropriately and rigorously? 

Reviewer #3: (No Response)

4. Have the authors made all data underlying the findings in their manuscript fully available?

Reviewer #3: (No Response)

5. Is the manuscript presented in an intelligible fashion and written in standard English?

Reviewer #3: (No Response)

6. Review Comments to the Author

Reviewer #3: (No Response)

7. PLOS authors have the option to publish the peer review history of their article (what does this mean?). If published, this will include your full peer review and any attached files.

Reviewer #3: **Yes: **Fabrizio D'Ascenzo

---

## [Editor Report · Acceptance letter]

2 Dec 2021

PONE-D-20-25643R2 

Incidence and predictors of mortality among persons receiving second-line tuberculosis treatment in sub-Saharan Africa: A meta-analysis of 43 cohort studies 

Dear Dr. Edessa:

I'm pleased to inform you that your manuscript has been deemed suitable for publication in PLOS ONE. Congratulations! Your manuscript is now with our production department. 

Kind regards, 

on behalf of

Dr. Olivier Neyrolles 

Section Editor

PLOS ONE